# Exploring Religiousness and Hope: Examining the Roles of Spirituality and Social Connections among Salvadoran Youth

**Pamela Ebstyne King [1],*, Jennifer Medina Vaughn [1]** **, Yeonsoo Yoo [1], Jonathan M. Tirrell [2], Elizabeth M. Dowling [2], Richard M. Lerner [2], G. John Geldhof [3], Jacqueline V. Lerner [4], Guillermo Iraheta [5], Kate Williams [5] and Alistair T. R. Sim [5]**

[1] Thrive Center for Human Development, Fuller Theological Seminary, Pasadena, CA 91101, USA; jloya@fuller.edu (J.M.V.); yeonsooyoo@fuller.edu (Y.Y.)

[2] Institute for Applied Research in Youth Development, Tufts University, Medford, MA 02155, USA; jonathan.tirrell@tufts.edu (J.M.T.); Elizabeth.dowling@tufts.edu (E.M.D.); Richard.lerner@tufts.edu (R.M.L.)

[3] Human Development and Family Sciences, Oregon State University, Corvallis, OR 97331, USA; john.geldhof@oregonstate.edu

[4] Lynch School of Education and Human Development, Boston College, Chestnut Hill, MA 02467, USA; Jacqueline.lerner.1@bc.edu

[5] Compassion International, Colorado Springs, CO 80921, USA; giraheta@compassion.com (G.I.); kwilliams@us.ci.org (K.W.); alsim@us.ci.org (A.T.R.S.)

**\*** Correspondence: pamking@fuller.edu; Tel.: +1+626-584-5320

**Abstract:** Given the strong link between religiousness and hope, we sought to further understand the relations of these potentially powerful resources for youth living in adversity. Although existing research suggests that religiousness might be associated with adolescent hope via spirituality and social connections, few studies have tested models that integrate both. Thus, as applied psychologists, the aim of this paper was to test a theoretical model in the lives of youth. Drawing on a Relational Developmental Systems metatheory, we sought to further elucidate the relations between religiousness and hope and to explore how involvement in the faith-based youth-development organization, Compassion International (CI), might facilitate character strengths like hope. In order to do so, we tested whether religiousness was directly and indirectly (via spirituality and social connection) related to hopeful future expectations, using a sample of 9–15-year-olds in El Salvador (M = 11.6 years; n = 888), half of whom were involved in CI and the other half of whom were a locally matched counterfactual sample. Structural equation models revealed that higher levels of religiousness were directly and indirectly associated with higher levels of hope in relation to higher levels of spirituality and social connections among these youth. CI-supported youth reported significantly higher levels of religiousness than the counterfactual sample. Findings suggest that the relationship between religiousness and hope is best understood when it incorporates youth's spirituality and social connections associated with religion.

**Keywords:** religiousness; transcendence; fidelity; spirituality; social connections; hope

## 1. Introduction

Many Salvadoran youth experience chronic economic inequality and a constant threat of violence associated with gang activity (Organisation for Economic Co-operation and Development 2014; UNICEF 2017). Despite living in these contexts, many youth are not limited to day-to-day survival but rather, move towards positive development and meaningful contributions to the community (Raposa

et al. 2018). Such findings have spurred interest in identifying personal strengths and optimizing environments that support youth thriving in the face of such adversity. While there are several strengths present within these youth, one character strength that has been identified as associated with positive development is hope (Callina et al. 2014); and one context that is accessible to most Salvadoran youth (De la Torre et al. 2014), is recognized to be a buffer against risk, and that promotes thriving is religion (see Hardy and King 2019 for review). Given the strong link between religiousness and hope (Ciarrochi and Heaven 2012; Hood et al. 2009), we sought to further understand the relations among these potentially powerful resources for youth living in adversity. Consequently, the aim of this paper was to test a theoretically predicated model in the lives of actual youth. Drawing on a Relational Developmental Systems meta-theory (Lerner et al. 2018; Lerner et al. 2015), which emphasizes that human development occurs through the bi-directional relationships between persons and the contexts (e.g., social, cultural, religious) in which they live, we explored the relationships between religiosity, spirituality, social connections, and hope.

Whereas existing research suggests that religiousness might be associated with adolescent hope via spirituality (Harley and Hunn 2015) and social connections (Stoddard et al. 2011a), few studies have tested models that integrate both spirituality and social connections, especially among youth living in chronic adversity, such as those in El Salvador. Thus, the aim of this paper was twofold (1) to further elucidate the relations between religiousness and hope, and (2) to explore how involvement in the faith-based, child-sponsorship organization, Compassion International (CI), was associated with religiousness. Specifically, we examined how religiousness, personal spirituality, and social connection were associated with hope among youth enrolled in CI programs and compared them to youth not supported by CI.

## 2. El Salvador and Compassion International (CI)

El Salvador is considered to be one of the most violent countries due to widespread gang activity, political instability, and poverty in the region (Chavez 2007). It has consistently had one of the highest homicide rates for young people in the world (UNICEF 2017). This reality is largely the product of a conflict-ridden and disastrous history of events whose consequences still exist today. Significant contributors to El Salvador's history include a 12-year armed conflict which resulted in over 750,000 deaths (United Nations 1993), followed by a devastating 7.6-magnitue earthquake in 2001. Pervasive violence and poverty continue within the country due to an upsurge of gang activity, rapid urbanization, and social inequality (Fay and Laderchi 2005). Nonetheless, the experience of adversity and violence do not inevitably thwart positive development and thriving. Given the relationship between hope and positive youth development (Schmid et al. 2011), understanding how hope may be part of this ecology has the potential to shed further light on possible resources to promote thriving in adversity.

The current study is part of the larger Compassion International Study of Positive Youth Development (Tirrell et al. 2019b, 2019c). Compassion International (CI) is a faith-based child-sponsorship organization that provides enrichment opportunities in local communities across 25 countries for children living in extreme poverty. CI is committed to alleviating poverty and promoting thriving by helping youth to capitalize on their strengths and aligning them with contextual resources, while recognizing the importance of religion and spirituality in their lives. By partnering with local churches and projects, CI provides funding and programming that is adapted locally and is tailored to the needs and strengths of each specific community, while emphasizing religion and spirituality as key assets (King et al. 2019; Sim and Peters 2014; Tirrell et al. 2019c). Activities in which the youth might be engaged in, include team sports, arts and crafts, trade skills workshops, and life-skills building workshops. In addition, the youth have the opportunity to participate in faith-based activities, including scripture reading, spiritual retreats, and worship. CI youth are also connected to individual, family, or group sponsors who support youth through monthly financial support, personal letters,

and in some cases, gifts, and personal visits.[1] CI emphasizes the importance of relationships between youth participants and caring adults (e.g., project directors, pastors, and mentors) (Glewwe et al. 2018; Sim and Peters 2014; Tirrell et al. 2019a). A recent study suggests that CI-involved youth benefit from this strength-based approach and exhibit significantly higher levels of hope (Glewwe et al. 2018). The researchers suggested that it is this holistic approach, which operates on the spiritual, psychological, and social fronts, that might be more likely to have positive effects on development, as opposed to economic sponsorship alone.

## 3. Hope

Within the psychological literature hope is conceptualized as a character strength or virtue. Hope is considered to include agency thinking that involves being confident in such a way that enables one to forward, and finding the will to do so (Snyder 1999). In reviewing various social theorists' conceptions of hope, Snow (2018) identified both between "agentic" forms of hope that stem from experiences of effectiveness in attaining goals and "receptive" hope that is inspired by the presence, example, narratives, or support of other(s). Within the PYD literature, Callina et al. (2017) identify hope as a character strength important for organizing and energizing behavior toward a productive adulthood. Callina et al. (2015) emphasize a multidimensional approach to hope by operationalizing it as a combination of positive future expectations, intentional self-regulation skills, and connectedness. Positive future expectations involve personal motivations that energize a young person's behavior in the direction necessary for meeting personal goals. Hope involves cognitive and emotional activation that motivates youth to set goals, regulate behavioral choices, and feel confident in their abilities. Social connections, the relational component of hope, link intentional self-regulation skills, and positive future expectations by providing youth with self-confidence and creating new pathways toward goal attainment. Within the context of the Philippines, Bernardo's (2010, 2015) research points to the importance of relationality for understanding hope. He demonstrated that people have internal and external loci of hope and experience hope from experiencing confidence in themselves as well as in others, such as family, friends, and God. Common to all of these formulations of hope is a desire for something and the belief and confidence that one can attain it, except for Snyder (1999), who also discuss a relational dimension of hope.

## 4. Religiousness and Direct Pathway to Hope

Religiousness refers to one's engagement with the doctrines, beliefs, and community of one's religious tradition (King and Boyatzis 2015; Pargament et al. 2013) and has long been considered a source of hope. Religious doctrines often provide a vision for a hopeful future, whether in the present life or the afterlife (Bennett 2011; Ciarrocchi et al. 2008; King et al. forthcoming b). Generally, theistic religions affirm a sovereign God and the religious community as sources of hope and support. Religious practices, such as prayer, scripture reading, meditation, and participation in a religious community are offered as a means of cultivating hope. Religious scholars and practitioners (Huuskes et al. 2016; Scioli 2010) and social science researchers (Callina et al. 2018; Ciarrochi and Heaven 2012; Gallagher and Lopez 2017; King et al. forthcoming b; Scioli 2010) emphasize the importance of religiousness for hope, and demonstrate their positive association.

## 5. Indirect Pathway

Although the relation between religion and hope has long been established, few investigations have examined the potential underlying processes that link them—especially among youth exposed to poverty and adversity. Research suggests that there are both personal and ecological assets available

---

1 See www.compassion.com for more information.

within religiousness that might promote the development of hope (Hardy et al. 2019; Scioli 2010; Scioli et al. 2011). Specifically, research documents the relation between religiousness and spirituality, as well as between religiousness and social connections. In addition, both spirituality and social connections have been demonstrated to be related to hope (see Figure 1).

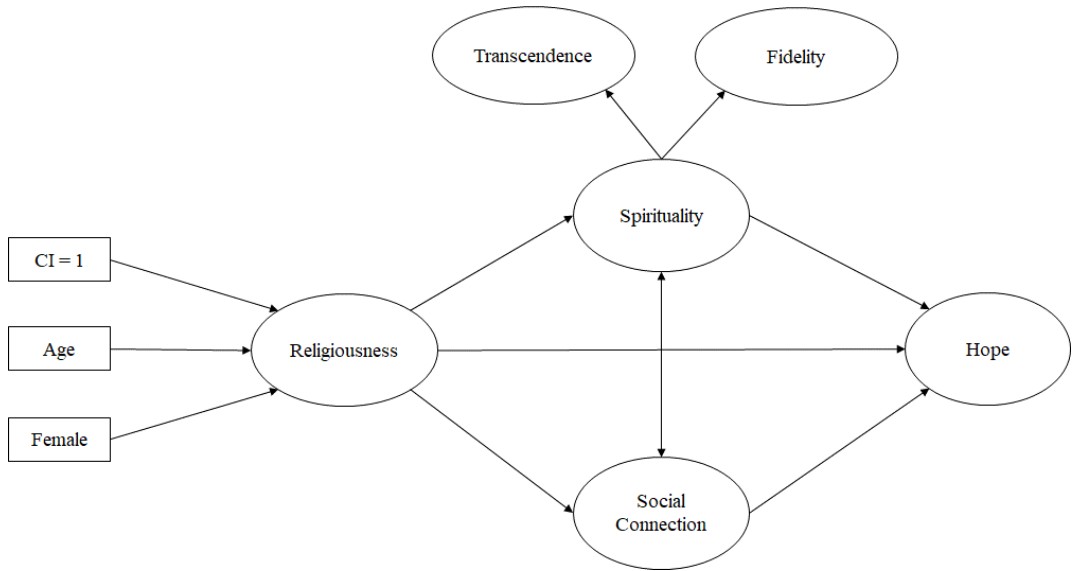

**Figure 1.** Hypothesized model.

*5.1. Religiousness, Spirituality, and Hope*

5.1.1. Religiousness and Spirituality

Religiousness is highly associated with spirituality, so much so that the distinction between the two constructs is often blurred (King and Boyatzis 2015; Pargament et al. 2013). Although they are overlapping constructs, spirituality is differentiated from religiousness. Spirituality involves the psychological processes at an individual level that involve transcending the self and growing in awareness and connection to a sense of the divine or of ultimacy that informs one's beliefs, identity, and actions (King and Boyatzis 2015; King et al. 2014; Lerner et al. 2003; Pargament et al. 2013). As such, spirituality might occur inside or outside of religious traditions. With increasing efforts to understand the nature and function of spirituality, two distinct dimensions of spirituality have emerged within the literature—transcendence and fidelity, which reflect one's experience of and response to their understanding of the divine or ultimacy (King et al. 2014, 2019; Kim et al. forthcoming). Specifically, transcendence is defined as an individual's personal awareness of and connection to a source of ultimacy beyond the self (i.e., God, Allah, the world, etc.) that provides meaning to their lives (King et al. 2019). Fidelity, then, involves an individual's response to transcendence, namely the shaping of their beliefs, identity, and purpose in a manner that spurs action oriented toward the common good (King et al. forthcoming a).

Although researchers have not examined transcendence and fidelity among Salvadoran youth, specifically, the high rates of religious affiliation in the country (Pew Research Center 2014) suggest it is an important area for research. Not only do Salvadoran youth consider religiousness and spirituality to be very important in their lives, but they also exhibit high levels of religious activity and engagement (Santacruz Giralt and Carranza 2009). Accordingly, while Salvadoran youth might access resources in other contexts, the current study highlights religiousness and spirituality as a noteworthy source. In the current study, spirituality is a latent factor comprised of both transcendence and fidelity.

### 5.1.2. Spirituality and Hope

Operationalizing spirituality as transcendence and fidelity highlights how spirituality is a resource for hope. Transcendence involves not only the cognitive awareness of something beyond the self and mundane, but also the emotional connection necessary to compel commitment to beliefs and ideals. Spirituality provides a transcendent source of ultimacy or meaning, which in the case of El Salvador is generally the Christian understanding of God, and an explicit transcendent worldview, including beliefs, values, and morals offered through both Catholic and Protestant traditions. Thus, a religious context offers a source of hope (e.g., God) and hopeful beliefs.

Spirituality not only involves the experience of transcendence, but it also involves a response to it and results in fidelity, which includes the processes of experiencing God, making meaning out of transcendent experiences and religious beliefs, and incorporating them into one's identity (King et al. 2014; Lerner et al. 2003). The integration of transcendent beliefs into one's identity has been demonstrated as effective for influencing how one interprets the world and finds meaning (Furrow et al. 2004; McAdams and McLean 2013; Schnitker et al. 2019). Thus, youth who are exposed to religion and adopt a hopeful narrative based upon their spiritual convictions might be more likely to filter their experiences through a hopeful lens and cope more effectively with adversity, which might enable them to pursue their goals for the future (King et al. 2019). These processes, especially when supported by others, can encourage hopeful future expectations and ultimately help regulate behavioral choices and energize action toward goal attainment (Schmid and Lopez 2011).

Research affirms these theoretical assertions. Spirituality has been associated with elevated levels of hope among adolescents (Ciarrochi and Heaven 2012; Harley and Hunn 2015; Huuskes et al. 2016). Specifically, fidelity to spiritual beliefs and values is associated with higher levels of hope and optimism (Ciarrochi and Heaven 2012; Huuskes et al. 2016). Research has demonstrated that youth with fidelity internalize transcendent beliefs into their identity in such a manner that they are motivated to act out their belief systems accordingly (Furrow et al. 2004; King et al. 2014). Studies of the relations between transcendence and hope are more limited. In a sample of adults, having a sense of closeness to God was related to increased hope and optimism (Ciarrocchi et al. 2008), and others have demonstrated that feeling loved by God was associated with higher levels of hope in adults (Hood et al. 2009). Other research has demonstrated that religion provides transcendent narratives that provide meaning and motivation for cultivating virtues like hope (Schnitker et al. 2019). In summary, through being aligned and connected with a loving God (transcendence) and being committed to those beliefs (fidelity), spirituality can be a resource for hope-filled narratives and beliefs reinforced by rituals, spiritual practices, and social relationships to promote hope.

### 5.2. Religiousness, Social Connection, and Hope

Of late, psychologists have endeavored to examine the multifaceted nature of religiousness and have identified the social benefits of engaging youth in religion (for reviews see Hardy et al. 2019; King and Boyatzis 2015). For most, religiousness is often accompanied by a religious community that provides the opportunity to engage in social relationships. Through involvement and investment in a religious community, youth have opportunities for constructive time spent with family, peers, and caring adults (Aoki et al. 2000; Dollahite and Marks 2018; Roehlkepartain and Patel 2006; Schwartz et al. 2006). These contexts also provide access to formal or informal mentorships in a uniquely intentional manner. In addition, these contexts might help meet an adolescent's desire for a sense of belonging and bonding (King 2003). Research demonstrates that religiously engaged youth report higher levels of social capital and social support than the less religious youth (Glanville et al. 2008; King and Furrow 2004; Muller and Ellison 2001). Similarly, in a sample of spiritual exemplars, 28 out of 30 youth emphasized connection to others as important in their experience of spirituality (King et al. 2017a). This research suggests that spirituality, often conceptualized as a more personal experience of the divine, involves and encourages social connection. Research also suggests that religious youth's positive networks extend beyond their religious congregation. For example, youth with higher religiousness

also report higher levels of positive peer interaction and trust with their closest friends (King and Furrow 2004). Furthermore, friendships that include comfortable discussions about spirituality with peers are associated with higher levels of spirituality in adolescents (Desrosiers et al. 2011), suggesting a bidirectional relation between spirituality and social connection.

The relational assets associated with religiousness potentially provide increased opportunities for social connections that encourage hope. Religious communities provide exposure to and interaction with individuals who teach about hope, model and exemplify hope, and provide the support necessary to practically pursue what one hopes for. Research indicates that religious youth report having caring adults who provided practical support that enables them to pursue their goals and achieve their dreams (King et al. 2014; Liang and Ketcham 2017).

These studies affirm the importance of considering extended conceptualizations and experiences of hope. In fact, Callina et al. (2018) emphasize social connection in their operationalization of hope, explaining that relationships are necessary to provide youth with confidence, inspiration, and collaborative means of pursuing goals for the future. This conceptualization is especially prudent when examining hope within traditionally collectivistic cultures such as El Salvador. Research by Bernardo (2010, 2015) provides sensitivity to these collectivistic tendencies by suggesting that hope, and influences upon it, can be separated into internal and external dimensions. The external dimension, which includes family, peers, and spiritual (or supernatural) beings aligns most closely to collectivistic understandings of hope (Bernardo 2010). He found that individuals who identify as more collectivistic have higher levels of external sources or loci of hope compared to those who strongly identify as individualistic. In other words, some people conceptualize hope as involving others or as co-participants in goal attainment. This conception of hope aligns with Snow's (2018) observation of "receptive" hope across different social theories, in which an individual experience hope that is "received from" or "inspired by" external sources and empowers agency. From this perspective, young people's experience of hope is derived from and supported by conjoint forms of agency that relate to connections with others.

Other research has demonstrated that connections to positive individuals can facilitate the development of hope and serve as a protective factor in the lives of young people, even within stressful environments (Resnick et al. 1997, 2004; Stoddard et al. 2011a; Wickrama and Bryant 2003). Similarly, findings that parent–family connectedness, school connectedness, and interaction with caring adults are associated with the development of hope among young people suggest that hope can be learned and nurtured (Resnick et al. 1997; Stoddard et al. 2011b). These connections bolster youths' confidence, inspire their beliefs in a hopeful future, and encourage them to take tangible steps towards their futures (Oyserman and James 2009). The collectivistic tendencies in El Salvador and the existing research confirming the association between religious engagement and caring relationships, underscore the need to explore the ecological assets within religiousness in the lives of Salvadoran youth, which might facilitate the development of hope.

In summary, the existing literature indicates that religiousness is associated with hope. Research clearly links religiousness with spirituality and also with social connections. In addition, there is evidence that spirituality and social connections are linked to hope. However, there are some noticeable gaps in the literature. First, few investigations have examined the complexity of the potential underlying social and spiritual psychological processes linking religiousness and hope during early adolescence, especially as they relate to positive development in the Salvadoran youth. Second, few studies have included multidimensional measures of spirituality and examined the role of social connections in a manner that would allow for the consideration of potential distinct pathways linking aspects of being religious to virtues like hope. To address these gaps, we tested whether general religiousness (i.e., attending religious services and valuing being religious and spiritual) was directly or indirectly associated with hope via spirituality and social connections among Salvadoran youth that are involved in church-based CI programs. Further, to determine whether there were meaningful differences between youth involved in CI programming, which is intentionally strengths-based and

religiously oriented, and a counterfactual sample of youth not enrolled in CI, we examined whether CI enrollment is associated with youth religiousness. We hypothesized that higher levels of religiousness would be related to higher levels of hope and that these associations would be explained by spirituality and social connections, given that research has suggested some evidence that involvement in CI programs, as well as age, and gender are closely related with adolescent religiousness (e.g., King and Boyatzis 2015; Sim and Peters 2014; Smith and Denton 2005). For example, based on previous research on adolescence, females as compared to males, report that religion is very important in their lives (King and Boyatzis 2015; Pew Research Center 2016). Studies have reported that rate of religiousness tend to decrease from childhood through adolescence (Bridges and Moore 2002; King et al. 1997). Accordingly, we anticipated that CI status, age, and gender would directly predict religiousness and be indirectly related to hope among Salvadoran youth.

## 6. Materials and Methods

Given the existing research that links religion and hope, the current study aimed to further understand the complexity of the relation between religion and hope among a sample of Salvadoran youth involved in faith-based programming through Compassion International (CI). We tested a conceptual model of hope by examining the relations among religiousness, spirituality, social connections, and hope, among a sample of Salvadoran youth. In addition, we included the CI-enrollment status (1 = enrolled), age, and gender (1 = female) as covariates in the study.

### 6.1. Participants

Participants were 888 Salvadoran youth derived from the first wave of the Compassion International (CI) Study of Positive Youth Development (PYD) (Tirrell et al. 2019c). Half of our sample were youth engaged in CI programs and the other half were a locally matched counterfactual sample. Participants' ages ranged from 9 to 15 years (M = 11.6 years, SD = 1.7) and 50% were female. Participants reported family religion as 72.2% Protestant Christian (Evangelical, Presbyterian, Lutheran, Anglican), 16.3% Catholic, 1.1% Adventist, and 10.4% reported no family religion. Participants were sampled from 20 communities that had local CI-supported project sites within urban and rural areas (70.4 % urban). CI-supported youth comprised 49.9% of the sample (n = 443), who were eligible to enroll in CI-supported programs based on multiple criteria including age, household monthly income and number of dependent, lack of any other outside sponsorship, and proximity to a CI-supported project site. The non-CI supported group were selected based on CI's eligibility criteria, including age, gender, and specific indicators of poverty at the time the CI-supported group were registered into the program.

For CI-supported youth, 20 CI project sites were selected from urban and rural locations to collect data. The project sites were selected by CI staff based on having strong program outcomes (e.g., graduation rates, program activities). Non-CI supported youth were recruited following a series of regional meetings with project leaders and school leadership in elementary and primary schools located in the same communities as CI project sites. School leaders then provided invitations and consent forms to distribute to participants and their parents. Youth were selected to participate in this study if they met the eligibility for CI involvement, but were not actually supported by CI. These eligibility criteria were based off indicators of poverty, including income, dependents in the home, and condition of the home. Identity of CI participants and counterfactuals were masked from CI and the research team. Data collection by independent data collectors took place in December 2016. Using electronic tablets, the data collectors read the questions to the participants and entered the responses for the child. The survey took approximately 30–45 minutes to complete. All data were de-identified.[2]

---

[2] The research team that was not based in El Salvador was given de-identified data that were collected by independent data collectors supervised by the CI regional staff. As such, the Institutional Review Board (IRB) of the research team's institution

## 6.2. Measures

We used multiple Likert-scale measures to explore the bivariate relations among religiousness, hope, spirituality, and social connections (see Tirrell et al. 2019c for initial measurement model). Religiousness was measured by a three-item self-reported questionnaire designed to assess the frequency of religious event participation, and personal importance and belief in religion and spirituality.[3] The first item asked the frequency of participation in religious events, with response options ranging from 1 = Never to 5 = More than once a week. Second, participants were asked to report how spiritual they are, with response options ranging from 1 = Not spiritual through 5 = Very spiritual. The last item asked participants how important being spiritual was to them. The response options ranged from 1 = Very important to 5 = Not very important and were reverse coded so that higher scores meant higher religiousness.

Youth spirituality was assessed using the Measure of Diverse Adolescent Spirituality (MDAS), which included items pertinent to 'transcendence' and 'fidelity' (see King et al. 2017b, 2019; Tirrell et al. 2019c). Consistent with the initial development and testing of the MDAS (see King et al. 2017b), transcendence was operationalized as a connection to something beyond the self, which might include an experience of the divine or supernatural other (i.e., God, Allah, Absolute Truth), family, friends, all of humanity, or nature. Fidelity was operationalized as clarity and conviction of beliefs that engage the young person in the world beyond the self. For the current study, four items each were used for transcendence and fidelity, respectively (see Table 1).[4] Adolescents indicated how true each statement was in their lives. Response options for all items used a 5-point Likert-type response ranging from 1 ("Not true in my life") to 5 ("Almost always true in my life"). The translation of the MDAS into Spanish, which is the official language of El Salvador, was done using a translation–back-translation procedure by different bilingual individuals in each country. The original publication of the MDAS describes the steps for "deep contextualization" of items into another language and culture (see King et al. 2017b). Although the original items of the MDAS were developed in Spanish and received many iterations of feedback from youth and adults (parents, practitioners, and scholars) in Mexico and modified accordingly, Salvadoran data collectors and local CI staff reviewed and suggested slight modifications of items for use in the current context. The Cronbach's alpha for the eight items of spirituality was 0.73 in this sample (0.67 for transcendence; 0.66 for fidelity).

---

granted the project exempt status for secondary data analysis. The research team based in El Salvador is composed of CI staff, who follow the CI policy of adhering to a specific country's governmental IRB requirements, if any. If, as is the case in El Salvador, no such governmental requirements exist, CI requires that before youth are assessed, signed parental consent forms, as well as youth assent (if below the age of consent), be obtained. All youth were told that there are no penalties if they elect not to participate and, that they can decide not to answer any question, and might end their participation at any time, again without any penalties (Tirrell et al. 2019c for full details). Data were de-identified from participants.

[3] Within the context of El Salvador, valuing being "religious" (religioso) or describing oneself as "religious" conveyed being a clergy or "a religious." Through pilot-testing and cognitive interviews, we determined that asking youth about being spiritual (espiritual) was the most accurate means of inquiring about their religious faith commitments.

[4] Although eleven items from the original transcendence scale were initially published with youth data from Mexico (see King et al. 2017b), the MDAS has since been refined (using the criteria of robustness and parsimony) to eight items total for the two subscales. These eight total items were selected based on findings from Tirrell et al. (2019c).

**Table 1.** Standardized factor loadings for indicators with latent constructs in confirmatory factor analysis (CFA).

| Construct | Indicator | Standardized Factor Loading |
|---|---|---|
| Religiousness | Religious Event participation | 0.47 *** |
| | Importance of being spirituality | 0.33 *** |
| | How spiritual I am | 0.48 *** |
| Spirituality | How true are the following statements in your life? | |
| Transcendence | I find meaning in life when I feel connected with God. | 0.59 *** |
| | I marvel in front of nature and God's creation. | 0.55 *** |
| | I feel God's presence in my life. | 0.58 *** |
| | I feel that there is someone bigger than me (God) that is concerned for me. | 0.59 *** |
| Fidelity | I try to incorporate my religion or spirituality in every aspect of my life. | 0.46 *** |
| | My spiritual beliefs define the way I see the world. | 0.55 *** |
| | I face the obstacles and problems in life when I think that my life is part of God's plan. | 0.52 *** |
| | Religion or spirituality is a big part of who I am. | 0.66 *** |
| Connection | How much do you agree or disagree with the following statements? | |
| | I matter to my friends. | 0.34 *** |
| | I think I have good friends. | 0.29 *** |
| | The adults of my community listen to what I have to say. | 0.52 *** |
| | The adults of my community make me feel important. | 0.57 *** |
| | I feel useful in my family. | 0.59 *** |
| | I have a lot of good conversations with my parents. | 0.46 *** |
| Hopeful Future Expectations | Think about how you see your future. What are your chances for the following? | |
| | Having a happy family life | 0.51 *** |
| | To live wherever you want | 0.46 *** |
| | Be respected in your community | 0.56 *** |
| | Have trustworthy friendships | 0.54 *** |
| | Be healthy | 0.55 *** |
| | Be safe | 0.66 *** |

*** $p < 0.001$.

To assess the positive social connections with people and institutions, we used items of 'Connection' drawn from the Five Cs of PYD (Geldhof et al. 2014), derived from the 4-H Study of Positive Youth Development (Lerner et al. 2005). Six items were used to measure the feeling of connection, including the subscales of connection to family (two items), peers (two items), and community (two items) (See Table 1). Response format for four items of connection to family and community ranged from '1' = 'totally disagree' to '5' = 'totally agree', and for two items of connection to peers ranged from '1' = 'hardly ever true' to '5' = 'always true'. The Cronbach's alpha for the 'Connection' items was 0.66 in this sample.

Hope was measured using a modified version of the Hopeful Future Expectations (HFE) Scale (Schmid et al. 2011). Six items were selected based on cultural appropriateness and psychometrics (see Tirrell et al. 2019c) to assess the likelihood of obtaining something in the future (see Table 1). Items included expecting a happy family life, living wherever you want, being respected in your community, having trustworthy friendships, being healthy, and being safe. Response options ranged from '1' = 'very low' through '5' = 'very high'. The Cronbach's alpha for the HFE items was 0.71 in this sample.

*6.3. Data Analysis*

Structural equation modeling (SEM) was used to investigate the relations between adolescent religiousness and hope by exploring the role of spirituality and social connections. Youth's age, gender, and CI-involvement were included as covariates in the structural model. The SEM process included two steps of validating the measurement model and the structural equation model. First,

the measurement model was evaluated by conducting confirmatory factor analysis (CFA) to define latent variables with indicators. In the second step, the structural models were tested by applying path analysis to investigate the relationships among the latent variables of religiousness, spirituality, social connections, and hope, to develop a conceptual framework of hope. We fitted models using multiple goodness of fit indices, as recommended by Brown (2006), including the root mean square error of approximation (RMSEA), the comparative fit index (CFI), the Tucker–Lewis index (TLI), and the standardized root mean square residual (SRMR). For the RMSEA and SRMR, values less than 0.08, and ideally below 0.05, were used to indicate an adequate and reasonable fit to the data (Browne and Cudeck 1993; Hu and Bentler 1999; MacCallum et al. 1996). Values of 0.90 or greater, and ideally above 0.95, were used to indicate good model fits for the CFI and TLI (Hu and Bentler 1999; Kline 2005; Raykov and Marcoulides 2006). All statistical analyses were conducted using IBM SPSS Statistics software (Version 24.0) and Mplus (version 8.1; Muthén and Muthén 2018).

## 7. Results

Within our sample, item-level missingness was under 1% of the data, allowing us to assume that data were missing at random (MAR), which can be addressed by using full information maximum likelihood (FIML) defaulted in Mplus. To account for nesting in project sites or schools, we used the maximum likelihood estimation with robust standard error (MLR) methods and the TYPE = COMPLEX command in Mplus (version 8.1; Muthén and Muthén 2018), which provides robust standard errors, to account for non-independence of observations.

First, we tested a measurement model to confirm our factor structures factor analysis (CFA) for estimating the latent variables of religiousness, spirituality, social connections, and hope. For spirituality as a second-order latent variable, a two-factor solution was used to involve two first-order latent variables of transcendence and fidelity. Correlated errors were added between two observed variables (two items in each of family, peers, and community), based on their theoretical relatedness. The factor loadings on all our latent variables were statistically significant at $p < 0.001$ (see Table 1 for indicator loadings). The test of the model fit was good: $\chi^2$ (217) = 261.00, $p < 0.05$; RMSEA = 0.02 (90% CI [0.01, 0.02]); CFI = 0.98; TLI = 0.98; SRMR = 0.03.

Second, the structural model tested our conceptual framework of the relationships among religiousness, spirituality, social connections, and hope in Salvadoran youth. All hypothesized paths to explore the potential relations between religiousness and hope were included in our test of the full structural model, in order to test both direct and indirect relationships between religiousness and hope. Next, we trimmed non-significant paths so that only statistically significant paths are shown in the final model.

For direct paths, we tested a structural model in which the influence of religiousness was directly associated with Salvadoran youths' HFE. The model demonstrated good fit to the data: $\chi^2$ (50) = 84.11, $p < 0.01$; RMSEA = 0.03 (90% CI [0.02, 0.04]); CFI = 0.96; TLI = 0.95; SRMR = 0.03. Results indicated that higher levels of religiousness were associated with higher levels of HFE ($\beta = 0.230$, $p < 0.01$). None of the covariates (e.g., age, gender, CI-enrollment status) were significantly related to the latent variable of religiousness ($\beta = -0.08$, $\beta = -0.07$, $\beta = 0.05$, $p > 0.05$, respectively).

Regarding indirect paths, all hypothesized paths were tested, including the second-order latent variables of spirituality and the latent variable of social connections, in addition to the independent and dependent variables (see Figure 1). In the full model, results indicated that all paths in the hypothesized model were positively significant, except the path from religiousness to HFE and the factor correlation between spirituality and social connections ($\beta = -0.07$; $\beta = 0.12$, $p > 0.10$, respectively). We found that the relationship between religiousness and HFE covaried through spirituality and social connections (Figure 2). In other words, religiousness promoted youth's spirituality and social connections, which in turn, positively influenced HFE. The model revealed significant total indirect effects between religiousness and hope ($\beta = 0.30$, $p < 0.001$) via spirituality ($\beta = 0.19$, $p < 0.001$), and via social connections ($\beta = 0.11$, $p < 0.001$).

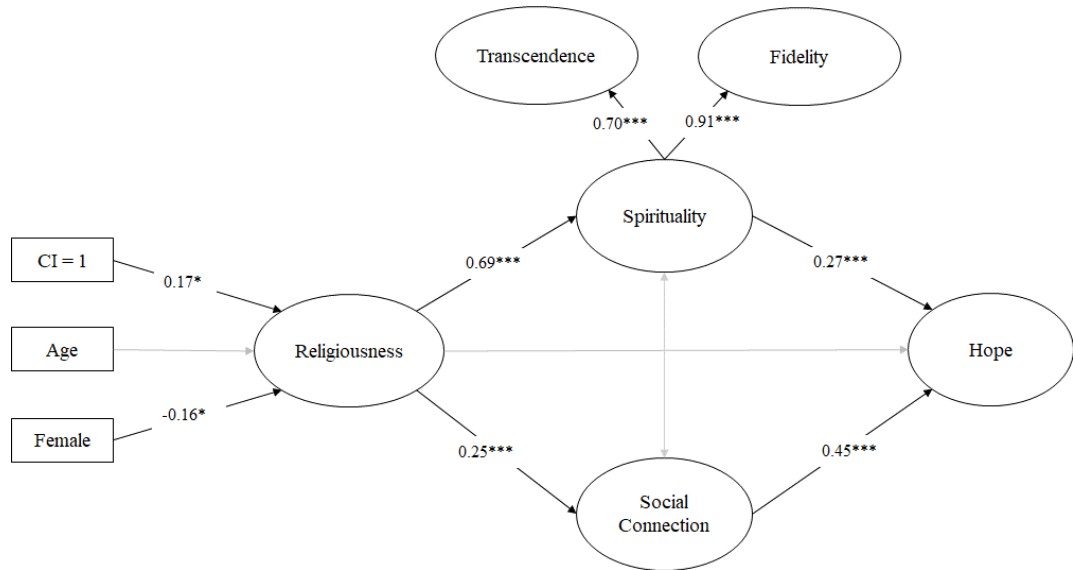

**Figure 2.** Final model. *** $p < 0.001$, * $p < 0.05$. Note: Regression coefficients are standardized.

The covariates of age, gender, and CI enrollment were also included in the model to test their association with the latent variable of religiousness. Results suggested that boys and CI-supported youth showed higher levels of religiousness ($\beta = -0.16$, $\beta = 0.17$, $p < 0.05$, respectively). After removing all non-significant paths, the re-specified model demonstrated good fit to the data: $\chi^2$ (266) = 350.76, $p < 0.001$; RMSEA = 0.02 (90% CI [0.01, 0.02]); CFI = 0.97; TLI = 0.97; SRMR = 0.04.

Correlations among the latent factors are shown in Table 2. Religiousness, spirituality, social connections, and HFE were significantly correlated. Especially, religiousness and spirituality were highly correlated ($r = 0.69$, $p < 0.001$). Although results presented non-significant relationship between spirituality and social connections in the final structural model, the latent factor correlations to test the statistical association among the latent variables showed a small but significant correlation between spirituality and social connections ($r = 0.17$, $p < 0.001$). Being female (girls) was negatively correlated with religiousness ($r = -0.08$, $p < 0.05$), indicating that males (boys) showed a higher level of religiousness. CI-enrollment were positively correlated with religiousness ($r = 0.08$, $p < 0.05$), indicating that the CI youth showed a higher level of religiousness.

**Table 2.** Correlations among the latent factors.

|  | Religiousness | Spirituality | Social Connections | Hope | Age | Female | CI-Enrollment Status |
|---|---|---|---|---|---|---|---|
| Religiousness | — |  |  |  |  |  |  |
| Spirituality | 0.69 *** | — |  |  |  |  |  |
| Social Connections | 0.25 *** | 0.17 *** | — |  |  |  |  |
| Hope | 0.30 *** | 0.34 *** | 0.49 *** | — |  |  |  |
| Age | 0.01 | 0.04 | 0.00 | 0.00 | — |  |  |
| Female | −0.08 * | −0.05 * | −0.02 | −0.02 | −0.00 | — |  |
| CI-Enrollment Status | 0.08 * | 0.05 | 0.02 | 0.02 | 0.05 ** | 0.01 | — |

*** $p < 0.001$, ** $p < 0.01$, * $p < 0.05$.

The results suggested that among these Salvadoran youth, higher levels of HFE were fully explained by spirituality and social connections being associated with religiousness. To determine whether spirituality and social connections influenced the relations between religiousness and HFE, we tested the indirect effect (Baron and Kenny 1986). Model results showed that the total effect was significant ($\beta = 0.24$, $p < 0.01$), but the direct effect was not significant ($\beta = -0.07$, $p > 0.10$). Results for the indirect effect from religiousness to HFE via spirituality and social connections were significant

($\beta = 0.21$, $p < 0.05$; $\beta = 0.09$, $p < 0.01$, respectively), indicating that a higher level of religiousness was associated with levels of HFE indirectly via spirituality and social connections.

## 8. Discussion, Limitations, and Future Research

The current study sought to further clarify the relations between religiousness and hope. We took the opportunity to explore the complexity of this topic with a group of youth living in adversity, in a culture that is highly religious, and in which half of the sample was involved in a faith-based organization—Compassion International (CI). Drawing on existing literature, we proposed, tested, and validated a conceptual model in which we posited that being religious would be relevant to hope when taking personal spirituality and one's social connections into consideration.

Given an increased awareness of the shortcomings of youth development programming that are based on deficit models and focus on the elimination of problems and disease (Lerner et al. 2015), global development organizations are seeking evidence-based approaches that are strengths-based and emphasize positive outcomes like becoming responsible and fulfilled adolescents and adults (e.g., UNICEF 2017; YouthPower Learning 2017). This study was part of the larger CI Study of PYD (see Tirrell et al. 2019b, 2019c) that was designed to understand how CI, a strengths-based and faith-based youth program, might promote positive development and thriving in young people living in poverty. The current study used cross-sectional data and, thus, we did not intend to evaluate programs or make any causational claims. Rather, our interests were exploratory and theoretical. We sought to further understand concurrent associations between religiousness and hope in order to gain insight into religion's potential connection to character strengths and virtues like hope. Our intention was to spur on further research that would effectively explore the roles of religiousness, spirituality, social relationships, and hope in order to eventually test causal pathways and provide practical guidance for effective youth programming in the future.

Consistent with existing research (Ciarrochi and Heaven 2012; Ciarrochi et al. 2007; DiPierro et al. 2018; Huuskes et al. 2016; Sabatier et al. 2011), we found that higher levels of religiousness were associated with higher levels of hope among these Salvadoran adolescents. When we incorporated spirituality and social connection into our analyses, we found that higher levels of religiousness were significantly associated with higher levels of personal spirituality and social connections which, in turn, were related to higher levels of hope.

Our findings provide additional evidence explaining how hope might relate to religion. Our model suggests that hope is associated with personal spirituality characterized by transcendence and fidelity. From this standpoint, having a sense of belonging, feeling that one matters to one's perception of the divine (in the case of this sample, God), and being committed to one's beliefs might be related to hope. Fidelity to beliefs might reflect personal internalization of hopeful religious narratives that orient a person's worldview towards a hopeful direction (King et al. 2019). Through their belief in a loving God and internalization of faith as a way of life, spirituality might promote hope beyond one's present circumstances.

In addition to identifying the spiritual aspects of religiousness that are related to hope, our study also pointed to the relevance of social connections associated with religious engagement to hope. As hypothesized, and consistent with other studies, our findings suggest that increased religious engagement was associated with higher levels of social relationships (King and Furrow 2004; Smith 2003). Our findings were also congruent with previous studies that demonstrated the association between relationships and having hope (Callina et al. 2014; Stoddard et al. 2011a). Although we did not measure connection in a manner that distinguishes between microsystems (e.g., family, peers, spiritual), our study yields further support for the importance of social relationships in potentially building and maintaining hope in young people.

Taken together, our findings inform the complex relationship between religiousness and hope and show that personal spirituality and social relationships associated with being religious are associated with having hopeful future expectations. This was the first study to systematically test the complex

association between religiousness and hope, including personal spirituality and social connections. Our findings point to the significance of having a sense of knowing and experiencing God, a commitment to spiritual beliefs and devoted faith, and a network of interpersonal relationships, as they relate to having hope amidst adversity. In this way, our study provides foundational evidence suggesting that religion and spirituality might be a source of hope in seemingly hopeless situations.

Consistent with previous research (Tirrell et al. 2019c), the CI-supported group demonstrated higher level of religiousness. Our findings reinforced the premise that youth involvement in faith-based CI programming impacted religious importance and participation among this sample of Salvadoran youth, which then indirectly influenced spirituality, social connections, and ultimately, hope. As one of the first studies to examine these constructs collectively in a group of Salvadoran adolescents involved in faith-based programming, this study suggests an important relation between hope in adversity and religious importance, participation, spirituality, and social connections.

There were a few discrepancies between our initial assumptions and results that are important to note. When considering the whole structural model, there was no significant intercorrelation between spirituality and social connections, which is inconsistent with other findings (Krause and Bastida 2011; Rew et al. 2004). One possible explanation for this finding is that the operationalized definitions or latent constructs of spirituality and social connections in past research are often general, using one or two items, that result in participants conflating spirituality and religion (Cotton et al. 2010; Zinnbauer et al. 1997), which involves communal or social experiences. We assessed a specific conceptualization of spirituality that aimed to capture an individual's experience of and response to transcendence (King et al. 2019). This finding points to the need for future clarification of the constructs actually being assessed by measures of "spirituality" to more clearly elucidate the complex psychological processes involved in spiritual and religious experience and the significance of distinguishing individual and communal experiences (see Hardy and King 2019).

In this study, gender, age, and CI-enrollment status were included as covariates in the analyses. Gender and CI-enrollment status were significantly linked to religiousness in the full model. Prior research has suggested that the level of religiousness differs by gender in adolescence—mostly, compared to boys, girls value and practice their religion more (Francis and Evans 1996). Surprisingly, results in the current study indicated that Salvadoran boys showed higher levels of religiousness. This might be related to gender inequities in El Salvador, wherein males often have greater access to religious resources, positions of leadership in religious contexts, and various experiences that might support the development of their religiousness.

This study has several limitations. First, longitudinal data are needed to fully test the influence of religiousness on hope via spirituality and social connections. Second, although we used a multidimensional assessment of spirituality, our assessment of hope was limited to hopeful future expectations. Future research would benefit from a more nuanced view of hope that moves beyond strict internal conceptualizations of individual goal attainment, towards a more comprehensive understanding that captures where hope comes from and how it is supported by external sources (Bernardo 2015; Snow 2018).

The bivariate relations established in our model point to directions for future research. The associations revealed in this model demonstrate the multifaceted nature of young people's experience and how it might be related to hope, which would benefit from longitudinal research and testing with diverse samples. Although variable-centered approaches such as ours provide theoretical clarity, they do not capture the unique experiences of individuals. The different elements of religious participation, salience, transcendence, fidelity, and relationships allow for scholars to begin to conceptualize how different youth might experience religion differently. As developmental science attempts to understand what kinds of resources promote positive development for different kinds of youth in different circumstances, this model highlights potential differentials (Bornstein 2017). Analyses that allow for ideographic findings or even person-centered analyses are warranted to further understand how these variables might function in unique persons and situations. Even within the context of the CI Study of

PYD, we have found that not all programs are created equal and not all programs are experienced the same by different youth (see Tirrell et al. 2019b).

The current study provides further insight into the complexity of hope for vulnerable adolescents in El Salvador. Given the extreme challenges facing youth in El Salvador and their capacity for thriving, we found that religiousness (comprised of valuing religion and attending religious events) played substantial roles in experiencing God, having spiritual beliefs and commitments, and accessing social networks that build hope. Although there are various resources within and outside of religious contexts that might enhance hope, our study provides compelling evidence for the importance of personal, transcendent, and social resources associated with religion that are related to hope for youth, especially those experiencing chronic adversity. All too often, youth live in desperate conditions that should preclude hope. Nonetheless, our study revealed conditions in which young people in these circumstances experience hope. Specifically, we found that youth experience hope when they have personal spirituality and social relations that are associated with engaging in religion. In this way, religion might be an important context that might provide the benefits of hope (e.g., source of transcendence, belief system, and people) and spirituality and relationships might be the ways in which young people engage in these ecological resources in a meaningful way that promotes hope. We anticipate that the theoretical clarifications suggested by our study will promote further research that provides additional insight into how to optimize the relationships between potential personal strengths, like hope, and existing ecological assets, like religion, in a way that enables youth to thrive.

**Author Contributions:** P.E.K., J.M.V., Y.Y., J.M.T., R.M.L., & J.V.L. conceptualized the study. Y.Y., J.M.T., & G.J.G. contributed to methodology and data analyses. E.M.D., G.I., K.W., & J.M.T. implemented the data curation. P.E.K., J.M.V., & Y.Y. wrote the manuscript. P.E.K., J.M.V., Y.Y., J.M.T., E.M.D., J.V.L., A.T.R.S. provided detailed editing of the full manuscript. All authors have read and agreed to the published version of the manuscript.

**Funding:** This research was funded by King Philanthropies and Compassion International, grant number CW2276870.

**Acknowledgments:** We would like to acknowledge Andrea Gonzalez of the Thrive Center and Heidi Johnson at for their administrative support.

**Conflicts of Interest:** The authors declare no conflict of interest.

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
