# Peer review of "Exploring Religiousness and Hope: Examining the Roles of Spirituality and Social Connections among Salvadoran Youth"

_religions, doi:10.3390/rel11020075_

Round 1

Reviewer 1 Report

Overall, this was one of the better papers I have read and/or reviewed this year.  The unique design offers several strengths and I commend the authors.

I offer several minor corrections and suggestions below:

Line 26 - Insert "inclined" before "towards" (or another similar word)

Line 102 - "have" should be "has" (subject/verb disagreement)

Line 152 - References need to be re-ordered in alphabetical order by first author's last name

Lines 170-171 - References need to be re-ordered in alphabetical order by first author's last name

Line 410 - a comma is needed after "e.g." 

Line 434 - Recommend dropping the s from "social connections"

Line 462 - I recommend dropping "actual"

Line 464-465 - Delete "to persists"

Line 479 - Delete "specific" ("Specifically" is used in the next sentence)

Line 482 - Delete "rich"

And now, one substantive point for the authors.  A real opportunity is missed to tell the reader about "CI."  It seems like a rather remarkable program, given the context and impact/influence on young lives. Yet, CI is defined and described only in quite general terms.  I would like to hear more about Compassion International throughout the paper. Please give details up front and then try to offer some explanations for the (positive) findings that are tied to specific aspects of the CI program in the Discussion.

Interesting work!

Author Response

Dear Reviewer,

Thank you for your review and valuable comments concerning our manuscript entitled “Exploring Religiousness and Hope: Examining the Roles of Spirituality and Social Connections among Salvadoran Youth” (Manuscript ID: religions-644913). Taking your suggestions and questions into consideration, we have substantially revised the manuscript. We hope you will agree that based on your suggestions that it is a stronger paper. Please find our responses to your queries below.

REVIEWER COMMENTS & AUTHOR RESPONSES:

 Overall, this was one of the better papers I have read and/or reviewed this year.  The unique design offers several strengths and I commend the authors.

I offer several minor corrections and suggestions below:

Line 26 - Insert "inclined" before "towards" (or another similar word)

Line 102 - "have" should be "has" (subject/verb disagreement)

Line 152 - References need to be re-ordered in alphabetical order by first author's last name

Lines 170-171 - References need to be re-ordered in alphabetical order by first author's last name

Line 410 - a comma is needed after "e.g." 

Line 434 - Recommend dropping the s from "social connections"

Line 462 - I recommend dropping "actual"

Line 464-465 - Delete "to persists"

Line 479 - Delete "specific" ("Specifically" is used in the next sentence)

Line 482 - Delete "rich"

Response: All formatting changes above have been completed in the manuscript as recommended.

And now, one substantive point for the authors.  A real opportunity is missed to tell the reader about "CI."  It seems like a rather remarkable program, given the context and impact/influence on young lives. Yet, CI is defined and described only in quite general terms.  I would like to hear more about Compassion International throughout the paper. Please give details up front and then try to offer some explanations for the (positive) findings that are tied to specific aspects of the CI program in the Discussion.

Response: We added a further explanation of the mission and programming of Compassion International and the activities that CI provided for youth in El Salvador as suggested.

Reviewer 2 Report

This study attempts to relate the religious beliefs and spirituality of
young people who participate in group activities and their influence on
a hopeful vision. It is interesting to be able to assess the influence
of the beliefs and spirituality of adolescents in their attitude towards
life and the world, more taking into account the social context in which
they find themselves and the possible preventive effect of this type
of activities. This work could be improved by expanding the description of the
population. It would be advisable to know sociodemographic
characteristics of both groups, family situation, as well as to
specify what type of activities were carried out within the Compassion
International (CI) and how long the adolescents had been attending
before the evaluation. Something fundamental in the investigation is to attend to the ethical
criteria, as the Helsinki criteria, but nothing appears in this regard
in the article. On the other hand, the possible influence on the hope of religiosity
or peer groups and the social support received within them is not
clear. It would be interesting to compare with adolescents who
belonged to some non-religious organization in order to take into
account the influence of social support. The results are confusing, such as not referring to the group to which
the data correspond (table 2), correlations with a qualitative
variable such as female gender (table 2), legends are missing in
Figure 2 and statements are made of results not presented (447 -448). It would be convenient to review the order of the authors in the
citations of the text (page 5-7).
The topic is interesting but ethical research standards must be met
and the results should be improved.

Author Response

Dear Reviewer,

Thank you for your review and valuable comments concerning our manuscript entitled “Exploring Religiousness and Hope: Examining the Roles of Spirituality and Social Connections among Salvadoran Youth” (Manuscript ID: religions-644913). Taking your suggestions and questions into consideration, we have substantially revised the manuscript. We hope you will agree that based on your suggestions that it is a stronger paper. Please find our responses to your queries below.

REVIEWER COMMENTS & AUTHOR RESPONSES:

This study attempts to relate the religious beliefs and spirituality of young people who participate in group activities and their influence on a hopeful vision. It is interesting to be able to assess the influence of the beliefs and spirituality of adolescents in their attitude towards life and the world, more taking into account the social context in which they find themselves and the possible preventive effect of this type of activities. This work could be improved by expanding the description of the population. It would be advisable to know sociodemographic characteristics of both groups, family situation, as well as to specify what type of activities were carried out within the Compassion International (CI) and how long the adolescents had been attending before the evaluation.

Response: This is an excellent point and a limitation of the current data set. Given being part of a larger project, this study was not designed as program evaluation. In this initial round of data, we did not collect programmatic information regarding family income and characteristics, other potential enrichment programs that the counterfactuals might be engaged in. Given that we do not have the detailed family and individual characteristics and the duration of involvement with programs. We recognize that these limitations would prohibit an effective evaluation of the CI program; however, given that our aim was to test a theoretical model on the entire sample in order to further understand the relationships between religiousness, spirituality, social connections, and hope, this information is not imperative. We highlight the significant finding that CI youth report higher levels or religiousness in order to highlight this distinction between the CI and non-CI groups and that it was a relevant variable to our model.

Something fundamental in the investigation is to attend to the ethical criteria, as the Helsinki criteria, but nothing appears in this regard in the article.

Response: Regarding the ethical criteria, we cited within section 2 which suggests that CI’s holistic approach is to support youth development, as opposed to economic sponsorship alone. In addition, we address ethical issues around data collection in section 2.1. It is noted in footnote 2 (page 7) “The research team that was not based in El Salvador was given de-identified data that were collected by independent data collectors supervised by CI regional staff. As such, the Institutional Review Board (IRB) of the research team’s institution granted the project exempt status for secondary data analysis. The research team based in El Salvador is composed of CI staff, who follow the CI policy of adhering to a specific country’s governmental IRB requirements, if any. If, as is the case in El Salvador, no such governmental requirements exist, CI requires that, before youth are assessed, signed parental consent forms and, as well, youth assent (if below the age of consent), be obtained. All youth were told that there are no penalties if they elect not to participate and, as well, that they can decide not to answer any question and may end their participation at any time, again without any penalties (Tirrell et al., 2019c for full details). Data was de-identified from participants.”

On the other hand, the possible influence on the hope of religiosity or peer groups and the social support received within them is not clear. It would be interesting to compare with adolescents who belonged to some non-religious organization in order to take into account the influence of social support.

Response: We agree. This is important to compare the effects of non-religious groups, social support, and hope. However, we do not have that data in the current data set. We will look to include such data in future data collections. 

The results are confusing, such as not referring to the group to which the data correspond (table 2), correlations with a qualitative variable such as female gender (table 2),

Response: We have added the explanation in Section 7 (page 9) regarding the correlations among the latent variables shown in Table 2.

legends are missing in Figure 2 and statements are made of results not presented (447 -448).

Response: Legends such as p-values exist in Figure 2. Regarding the statements on results, we revised the paragraph to clarify the indirect path model results shown in Figure 2 (Section 7, page 9).

It would be convenient to review the order of the authors in the citations of the text (page 5-7). 

Response: The order of the authors in the citations of the text were reviewed and some were corrected.